# Efficacy of US, MRI, and F-18 FDG-PET/CT for Detecting Axillary Lymph Node Metastasis after Neoadjuvant Chemotherapy in Breast Cancer Patients

**DOI:** 10.3390/diagnostics11122361

**Published:** 2021-12-14

**Authors:** Umit Turan, Murat Aygun, Berna Bozkurt Duman, Aygül Polat Kelle, Yeliz Cavus, Zeynel Abidin Tas, Ahmet Baris Dirim, Oktay Irkorucu

**Affiliations:** 1Adana City Research and Training Center, Department of General Surgery, Saglik Bilimleri University, Adana 01230, Turkey; murataygun27@hotmail.com (M.A.); drbarisdirim@yahoo.com (A.B.D.); 2Adana City Research and Training Center, Department of Medical Oncology, Saglik Bilimleri University, Adana 01230, Turkey; berboz@hotmail.com; 3Adana City Research and Training Center, Department of Nuclear Medicine and Molecular Imaging, Saglik Bilimleri University, Adana 01230, Turkey; dolunay_75@hotmail.com; 4Adana City Research and Training Center, Department of Radiology, Saglik Bilimleri University, Adana 01230, Turkey; ylzbyglu21@gmail.com; 5Adana City Research and Training Center, Department of Pathology, Saglik Bilimleri University, Adana 01230, Turkey; zeynelabidin46@gmail.com; 6Clinical Sciences Department, College of Medicine, University of Sharjah, Sharjah 27272, United Arab Emirates; oktaytip@yahoo.com

**Keywords:** axillary assessment, breast cancer, neoadjuvant chemotherapy, ultrasound, MRI, F-18 FDG-PET/CT

## Abstract

Background: The aim of this study was to investigate the efficacy of post-neoadjuvant chemotherapy (NAC) ultrasound (US), magnetic resonance imaging (MRI), and F-18fluorodeoxyglucose positron emission tomography (F-18 FDG-PET/CT) for detecting post-NAC axillary lymph node(ALN) metastasis in patients who had ALN metastasis at the time of diagnosis. Methods: This study included all breast cancer patients who received NAC for ALN metastasis; underwent axillary assessment with US, MRI, or F18FDG-PET/CT; and then were operated on in the General Surgery Clinic, Adana City Research and Training Hospital, Turkey. Patients’ data were recorded, including demographic data, clinicopathological parameters, NAC regimens, and operation types. The axillary response to chemotherapy on post-NAC US, MRI, and F-18 FDG-PET/CT was compared with the postoperative histopathological result of the ALN. Results: The study included a total of 171 female patients. The mean age of the patients was 53.28 ± 10.62 years. The post-NAC assessment revealed that the sensitivity, specificity, positive predictive value (PPV), and negative predictive value (NPV) of US for detecting ALN metastasis were 59.42%, 82.35%, 82.00%, and 60.00%, respectively, while the same measures regarding MRI for detecting ALN metastasis were 36.67%, 77.78%, 73.33%, and 42.42%, respectively. The sensitivity, specificity, PPV, and NPV of F-18FDG-PET/CT were 47.50%, 76.67%, 73.08%, and 52.27%, respectively. The evaluation of dual combinations of these three imaging techniques showed that the specificity and PPV of the combined use of US and F-18FDG-PET/CT was 100%. Conclusions: The results showed that US has the highest sensitivity and specificity for detecting ALN metastasis after NAC. Furthermore, ALND may be preferred for these patients instead of SLNB if both examinations simultaneously indicate lymph node metastasis in the post-NAC assessment with US and F-18 FDG-PET/CT. SLNB may be preferred if these two examinations simultaneously show a complete response.

## 1. Introduction

Breast cancer treatment requires a multidisciplinary approach, including surgery, systemic chemotherapy, and radiotherapy. Neoadjuvant chemotherapy (NAC), in general terms, is a preferred preoperative treatment modality for some early-stage and locally advanced breast cancer patients to increase survival rates, reduce the extent of the disease before surgery, turn inoperable tumours into operable tumours, and allow breast-conserving surgery instead of total mastectomy [1,2]. It is still controversial how post-NAC axillary management should be undertaken in patients with axillary lymph node metastasis before the NAC. Approximately 40% of breast cancer patients with axillary lymph node metastasis have a complete axillary response after NAC [3]. While in the past, axillary lymph node dissection (ALND) was the standard treatment after NAC for the patients with axillary lymph node metastasis, it has recently been reported that some patients with a complete axillary response after NAC can be treated with sentinel lymph node biopsy (SLNB) to avoid the morbidity of axillary dissection [3,4]. Therefore, the accurate evaluation of post-NAC axillary response is important in terms of determining the preferred type of operation for axillary management.

In addition to physical examination, ultrasound (US), magnetic resonance imaging (MRI), and F-18 fluorodeoxyglucose positron emission tomography (F-18 FDG-PET/CT) are most commonly used to evaluate post-NAC axillary response. Instead of ALND, SLNB may be preferred for patients with axillary lymph node metastasis who have received NAC and are considered to have a complete axillary response after NAC based on the findings of physical examination and these imaging techniques. For this reason, the reliability of these imaging techniques for post-NAC axillary assessment increases in importance. Studies have reported the sensitivity and specificity of US for evaluating post-NAC axillary response between 28.6–81.5% and 33.0–100%, respectively [5,6,7]. The sensitivity and specificity of MRI have been reported between 38.0–87.9% and 50.0–95.5%, respectively [5,6,8]. Studies on the efficacy of F-18 FDG-PET/CT after NAC have tended to focus on the evaluation of breast response. However, there are few studies on the efficacy of F-18 FDG-PET/CT for the evaluation of axillary response. Current studies have reported the sensitivity and specificity of F-18 FDG-PET/CT between 22.0–88.0% and 63.0–87.5%, respectively [9,10]. Regarding the positive predictive values (PPV) and negative predictive values (PPV) of these imaging techniques, previous studies have reported the PPV of US, MRI, and F-18 FDG-PET/CT between 57.0–84.0%, 29.4–93.0%, and 36.8–85.7%, respectively [8,9,10,11,12]. The NPV of these imaging techniques has been reported between 38.0–80.8%, 26.0–97.3%, and 28.0–94.4%, respectively [6,8,9,10].

The aim of this study was to investigate the reliability of US, MRI, and F-18 FDG-PET/CT for detecting axillary lymph node metastasis in the post-NAC assessment of axillary lymph nodes in breast cancer patients who had axillary metastasis at the time of diagnosis. In addition, the efficacy and reliability of dual combinations of these imaging techniques were investigated.

## 2. Materials and Methods

The study included all female patients who presented to the General Surgery Clinic of Adana Numune Training, the General Surgery, and Surgical Oncology Clinics of Research Hospital and Adana City Training and Research Hospital. The participants all had a diagnosis of invasive breast cancer between 1 January 2015 and 30 November 2020; received NAC for axillary lymph node metastasis; underwent assessment with at least one of the imaging techniques of US, MRI, and F-18 FDG-PET/CT after NAC; and underwent surgery. Included in the study were patients with clinically suspected axillary metastasis in the staging performed at diagnosis or with histopathologically confirmed axillary metastasis by core biopsy or fine-needle aspiration biopsy. Excluded from the study were male patients, patients with distant metastasis, those who did not receive NAC, those who received NAC but did not have any post-NAC imaging, and those who refused surgery results.

Patient data were analysed retrospectively from the hospital patient information system. Data were recorded, such as age; menopausal status; neoadjuvant chemotherapy regimen (docetaxel + doxorubicin + cyclophosphamide (TAC), TAC + trastuzumab, and others); tumour size; histopathological and immunohistochemical features of the tumour; axillary lymph node status; clinical stage (according to the American Joint Committee on Cancer (AJCC)); the response of axillary lymph nodes to chemotherapy on post-NAC US, MRI, and F-18 FDG-PET/CT scans; type of surgery (modified radical mastectomy(total mastectomy + ALND), total mastectomy + SLNB, lumpectomy + SLNB); and postoperative histopathological evaluation of axillary lymph nodes.

## 3. Imaging Techniques

All post-NAC axillary imaging techniques were evaluated in the presence of metastatic axillary lymph node (incomplete response) and absence of metastatic axillary lymph node (complete response). Patients with suspected lymph node metastasis in the assessment were included in the metastatic lymph node (incomplete response) group.

In the axillary assessment carried out with US, metastasis was evaluated based on the presence or absence of any of the morphological features, such as loss of fatty hilum in the lymph node, cortex thickness greater than 3 mm, irregular shape, and increased extrahilar vascularity. Axillary complete response was defined as the appearance of completely normal lymph nodes on US scans.

Breast MRI images of the study participants were evaluated from the hospital patient information system by a single radiologist experienced in this field. While identifying axillary lymph node metastasis in the MRI assessment, the determination of metastasis was made depending on the presence or absence of any of the following morphological features, such as increased contrast uptake of the lymph node, irregular appearance, size over 5 mm, and the absence of fatty hilum. Axillary complete response was defined as the appearance of completely normal lymph nodes on MRI.

An analogue PET/CT device was used in the study. Whole-body 18F-FDG PET/CT scans were obtained on a Biograph PET/CT system (Siemens Molecular Imaging, Hoffman Estates, IL, USA). The system consists of a full ring dedicated PET and a 2-slice spiral CT. F-18 FDG-PET/CT uptake with a higher intensity than that of normal soft tissue was considered positive for axillary lymph node metastasis on the F-18 FDG-PET/CT scans. The number of separate FDG-avid lesion in the breast and regional lymph node stations was visually assessed after anatomical localization. Any uptake in axilla that could be mapped to a lymph node was considered abnormal and was rated for metastasis (Figure 1). A given PET focus was rated as positive in a small lymph node rather than in a large lymph node, accounting for partial volume effects. Region of interest (ROI) was placed over the most intense area of 18F-FDG accumulation in axillary lymph nodes. All F-18 FDG-PET/CT images were analysed retrospectively by a single nuclear medicine physician.

## 4. Histopathological Examination

On histopathological examination, the excised lymph nodes were examined after formalin, paraffin, haematoxylin, and eosin procedures. The AJCC treatment response scoring was used to evaluate the presence of axillary complete response. While performing all post-NAC evaluations, the values determined to be a partial response were considered as residual tumours and were evaluated within the incomplete response group. Oestrogen receptor (ER) and progesterone receptor (PR) expression and HER2/neu status were evaluated in surgically excised samples using the standard avidin-biotin complex IHC staining. ER and PR receptor status were evaluated using the Allred scoring. HER2 staining was scored as 0, 11, 21, or 31 points. Tumours with a score of 31 points were classified as HER2-positive, while tumours with a score of 0 or 11 points were classified as HER2-negative. For tumours with a score of 21 points, the HER2 score was determined using fluorescent in-situ hybridization. Lymph nodes with micrometastases (<2 mm) were deemed metastatic.

The patients’ axillary lymph node statuses were designated either benign (complete response) or malignant (incomplete response). These designations resulted from postoperative histopathological evaluation along with analysis of post-NAC preoperative US, MRI, and PET, and combinations of these imaging techniques, including US + MRI, US + F-18 FDG-PET/CT, and MRI + F-18 FDG-PET/CT. Moreover, the specificity; sensitivity; negative and positive predictive values of the US, MRI, and F-18 FDG-PET/CT scans; and their dual combinations were calculated for post-NAC axillary assessment.

## 5. Statistical Analysis

Data are presented as mean (±standard deviation) and number (percentage). Sensitivity, specificity, PPV, and NPV were calculated for each imaging modality and for dual combinations of these imaging techniques. The efficacy of US, MRI, and F-18 FDG-PET/CT for detecting axillary metastasis after NAC was evaluated by receiver operating characteristic (ROC) analysis. IBM SPSS Statistics version 26.0 software was used for the analyses. A *p*-value less than 0.05 was considered statistically significant.

## 6. Results

Of the 191 patients recorded from the hospital patient information system, four were excluded from the study for being cases of male breast cancer, two for receiving hormonotherapy as neoadjuvant therapy, 12 for not having a post-NAC imaging examination, and two for refusing surgery after NAC. Thus, the data analysed consisted of a total of 171 female patients. It was found that 120 patients who completed NAC were evaluated with US, 48 patients were evaluated with MRI, and 140 patients were evaluated with F-18 FDG-PET/CT. The mean age of the patients was 53.28 ± 10.62 years. Among the study’s patients, 66 (38.8%) were in the premenopausal period, while 104 (61.2%) were in the postmenopausal period. Of the patients scheduled for NAC, 28 (16.4%) had stage IIA disease, 55 (32.2%) had stage IIB disease, 82 (48%) had stage IIIB disease, four (2.3%) had stage IIIB disease, and two (1.2%) had stage IIIC disease. Furthermore, 103 patients (60.2%) received TAC, while 41 patients (24%) received TAC + trastuzumab as their NAC regimen (Table 1).

The histopathological examinations revealed that 141 patients (82.5%) had invasive ductal carcinoma, and 14 patients (8.2%) had invasive lobular carcinoma. On immunohistochemical examination, 132 patients (77.2%) were ER-positive, and 123 patients (71.9%) were PR-positive. In terms of HER2 status, 49 patients (28.7%) had a score of 0, 27 patients (15.8%) had a score of 1, 44 patients (25.7%) had a score of 2, and 51 patients (29.8%) had a score of 3. The most common operation performed on these patients was modified radical mastectomy (*n* = 153, 89.5%) (Table 1).

In the post-NAC assessment of axillary lymph nodes with US, benign lymph nodes (complete response) were reported in 70 patients (58.3%) and malignant (incomplete response) lymph nodes in 50 patients (41.7%). Using MRI scans, benign lymph nodes were reported in 33 patients (68.8%) and malignant lymph nodes in 15 patients (31.3%). For evaluations via F-18 FDG-PET/CT, benign lymph nodes were reported in 88 patients (62.9%) and malignant lymph nodes in 52 patients (37.1%) (Table 2).

Postoperative histopathological evaluation of the axillary lymph nodes of the 171 patients included in this study revealed that 76 patients (44.4%) achieved a complete response with NAC. When evaluated with histopathological data, the sensitivity, specificity, PPV, and NPV of US for detecting axillary lymph node metastasis were 59.42%, 82.35%, 82.00%, and 60.00%, respectively, while the same measures using MRI for detecting axillary lymph node metastasis were 36.67%, 77.78%, 73.33%, and 42.42%, respectively. The sensitivity, specificity, PPV, and NPV of F-18 FDG-PET/CT for detecting axillary lymph node metastasis were 47.50%, 76.67%, 73.08%, and 52.27%, respectively (Table 3).

The evaluation of combining the imaging techniques showed that the US and F-18 FDG-PET/CT combination had a specificity of 100% and a sensitivity of 57.89%, while the sensitivity and specificity of the F-18 FDG-PET/CT and MRI combination were 40.00% and 91.67%, respectively. The sensitivity and specificity of the US and MRI combination were 41.18% and 84.62%, respectively. The use of the three imaging techniques together achieved a sensitivity of 50.00% and a specificity of 100% (Table 4). The evaluation of the PPV and NPV of the combinations of these imaging techniques revealed that the US and F-18 FDG-PET/CT combination had the highest PPV and NPV, with 100% and 60.00%, respectively. The PPV and NPV of combining the three imaging techniques are shown in Table 4.

The area under the ROC curve (AUC value) for each imaging technique and comparison of the different imaging techniques are shown in Figure 2. Evaluation of the techniques individually showed that the AUC values of US, MRI, and F-18 FDG-PET/CT were 0.697, 0.597, and 0.677, respectively, with US having the highest AUC value. Evaluation of the dual combinations of these examinations revealed that the AUC values of the US + MRI, US + F-18 FDG-PET/CT, and MR + F-18 FDG-PET/CT combinations were 0.681, 0.803, and 0.658, respectively, with the US + F-18 FDG-PET/CT combination having the highest AUC value among the dual combinations (Figure 3).

We also performed a subgroup analysis and excluded patients with SLNB. When patients who underwent only axillary dissection were analysed, the US’s sensitivity, specificity, PPV, and NPV values were 61.90%, 78.04%, 81.25%, and 57.14%, respectively. For MRI, these values were 39.28%, 75%, 78.57%, 34.61%, while for F-18 FDG-PET/CT, these values were 48.64%, 78.84%, 76,59% and 51.89% respectively. In addition, we determined sensitivity, specificity, PPV, and NPV values in combinations of these imaging methods. The values mentioned for the combined use of US and MRI were 50%, 85.71%, 88.88%, and 42.85%, while for the combined use of US and F-18 FDG-PET/CT, these values were 63.88%, 100%, 100%, and 59.37%, respectively. Additionally, for the combined use of MRI and F-18 FDG-PET/CT, the above-mentioned values were 42.10%, 100%, 100%, 45%, respectively.

Finally, these values were 58.33%, 100%, 100%, and 58.33%, respectively, when US, MRI, and F-18 FDG-PET/CT were used together.

## 7. Discussion

Surgical management of the axilla is still controversial in patients who had axillary lymph node metastasis at diagnosis and who achieve a complete axillary response after NAC. In the literature, there are studies recommending ALND as well as studies recommending SLNB for patients with a complete axillary response after NAC [3,4]. Therefore, it is very important to determine which patients will benefit from SLNB after NAC. For this reason, the efficacy of the imaging techniques is of great importance. The most important result of this study is that the combination of US and F-18 FDG-PET/CT examinations in the post-NAC axillary assessment can be effective and reliable for detecting axillary lymph node metastasis. This study demonstrated that the combined use of US and F-18 FDG-PET/CT for detecting axillary lymph node metastasis in breast cancer patients had a specificity of 100% and a PPV of 100%. In other words, the visualization of axillary metastasis on both US and F-18 FDG-PET/CT simultaneously suggests that all the patients had metastatic axillary lymph nodes. At the same time, the visualization of a complete axillary response on these two examinations is consistent with the histopathological result of a complete response in all patients. Based on this result, it may be more appropriate to prefer ALND instead of SLNB when the US and F-18 FDG-PET/CT combination indicates a malignant lymph node. Similarly, SLNB may be preferred for this patient group if the complete axillary response to NAC is simultaneously demonstrated by US and F-18 FDG-PET/CT. In their study on 139 patients, You et al. found a PPV value of 84% for the combined use of US and F-18 FDG-PET/CT for detecting axillary lymph node metastasis after NAC, which is as high as in the current study. However, they reported the specificity of this dual combination as 73% [9].

In this study, the evaluation of the combined use of US and F-18 FDG-PET/CT revealed a specificity of 100% and a PPV of 100%, which are slightly higher than the values found in previous studies. The reason for such a high PPV and specificity may be that suspected lymph nodes in the post-NAC assessment with both US and F-18 FDG-PET/CT were considered metastatic in this study. Moreover, lymph nodes with a maximum SUV above zero were considered metastatic in the assessment performed with F-18 FDG-PET/CT. This may have increased PPV and specificity values.

When these three examinations were evaluated individually in this study, US had the highest sensitivity and specificity values, with 59.42% and 82.35%, respectively. US also had highest PPV and NPV among these three examinations, with 82.00% and 60.00%, respectively. In their systemic review including 572 patients, Schipper et al. evaluated these three imaging techniques and, similar to the current study, found that US had the highest sensitivity and specificity rates for evaluating the complete axillary pathological response after NAC. In the same study, the PPV and NPV of US for evaluating the axillary complete response were reported as 57% and 71%, respectively.

Studies in the literature have reported sensitivity and specificity values between 54.00–95.50% and 57.33–87.90%, respectively, for the post-NAC assessment of axillary complete response with MRI [5,6,9,13]. In this study, the specificity of MRI was found to be 77.80%, which is in line with the literature. However, unlike previous studies, the sensitivity of MRI for detecting axillary complete response was found to be 36.67%. The reason for such a low sensitivity value may be the small number of patients evaluated with MRI after NAC.

Studies on post-NAC response assessment with F-18 FDG-PET/CT have mostly focused on breast response. The number of studies on axillary response is limited. In their study of 40 patients investigating the efficacy of early F-18 FDG-PET/CT examinations after the second cycle of neoadjuvant chemotherapy, Andrade WP et al. reported a sensitivity of 68% and a specificity of 75% for F-18 FDG-PET/CT [14]. Similarly, the study of Rousseau C. et al. of 64 patients to evaluate early response reported a sensitivity of 88% and a specificity of 73% for F-18 FDG-PET/CT. In this study, 120 patients were evaluated with F-18 FDG-PET/CT after NAC, and the sensitivity and specificity of this examination were found to be 47.50% and 76.67%, respectively. The reason for the low sensitivity here is that other studies compared the post-NAC F-18 FDG-PET/CT with the F-18 FDG-PET/CT performed at the time of diagnosis. Although the patients included in this study did have pre-NAC F-18 FDG-PET/CT results, evaluation here was only made between the post-NAC F-18 FDG-PET/CT and the histopathology results. In addition, an analogue F-18 FDG-PET/CT device was used in this study. Digital F-18 FDG-PET/CT devices provide better volumetric resolution, higher sensitivity, and quantitative accuracy than analogues, allowing for more accurate detection of small lesions. We think that our results would be better if digital F-18 FDG-PET/CT was used in our study.

We also performed subgroup analysis in our study. Patients who underwent SLNB were excluded from the study, and only the data of patients who underwent axillary dissection were analysed. Even in this subgroup, the US had the highest sensitivity and specificity for the post-NAC axillary assessment when the three most commonly used imaging techniques were compared. Furthermore, the evaluation of the dual combinations of these imaging techniques showed that US and F-18 FDG-PET/CT combination had a PPV of 100% and a specificity of 100%.

Previous studies have reported varying results regarding the efficacy of these three imaging techniques for post-NAC axillary assessment. The differences in these results can be explained by the methodology of the studies, racial differences, and variables in the clinicopathological characteristics of the patients included in the studies. The summary results of the previous studies are presented in Table 5 [5,6,7,8,9,10,11,12,13,15,16,17,18,19,20].

The key limitations of this study are its retrospective design and the relatively small sample size. In addition, Axillary biopsy was not required in patients who were clinically considered to have axillary metastases in the previous guidelines available at the time of the study. Likewise, clipping of axillary lymph nodes with metastasis was not routinely recommended because sentinel lymph node or targeted biopsy was not routine after neoadjuvant treatment at that time. On the other hand, in today’s perspective, the lack of clipping in patients with sentinel lymph nodes and the absence of axillary biopsies in patients with clinically axillary metastases are limitations and shortcomings of this study. Despite these drawbacks, the authors believe that this study is valuable in terms of comparing the three most commonly used imaging techniques for post-NAC assessment of axillary lymph nodes since the number of studies comparing them is extremely limited in the literature. Moreover, the authors are of the opinion that the PPV and specificity values of 100% for the combined use of US and F-18 FDG-PET/CT for post-NAC assessment of axillary lymph nodes is a valuable result in terms of clinical practice. This is because there are very few studies with similar features in the literature of combining these imaging techniques.

One possible reason why most patients in this study had undergone MRM after NAC may be that they have a preference due to their cultural characteristics. Other possible reasons include that patients do not accept the possibility of breast-conserving surgery after NAC or that the use of sentinel lymph node biopsy after NAC was not widespread in Turkey at the time of the study. However, although this may seem like a disadvantage, the fact that most of the patients had undergone MRM after NAC allowed the histopathological evaluation of a large proportion of the axillary lymph nodes, providing an important advantage for this retrospective study to collect data and perform accurate analysis.

In conclusion, US had the highest sensitivity and specificity for the post-NAC axillary assessment in this study when the three most commonly used imaging techniques were compared. Furthermore, the evaluation of the dual combinations of these imaging techniques showed that the combined use of US and F-18 FDG-PET/CT had a PPV of 100% and a specificity of 100%. Based on this result, it is theorised that ALND may be the right surgical choice for these patients instead of SLNB if the combined use of these two imaging techniques indicates metastatic lymph node in the axilla after NAC. Alternatively, SLNB may be the right surgical choice if a complete axillary response is identified on both these examinations. Although this study is encouraging, there is an urgent need to support these results with further clinical studies in order to clarify the way forward.

## Figures and Tables

**Figure 1 diagnostics-11-02361-f001:**
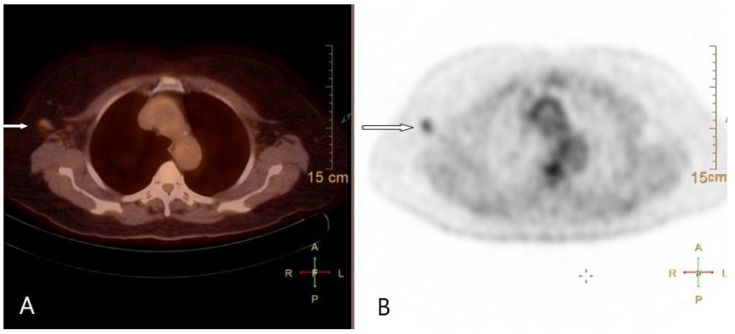
FDG PET/CT (**A**) and FDG PET (**B**) images shows FDG uptake (arrows) in a right axillary lymph node with a maximum standardized uptake value of 2.0.

**Figure 2 diagnostics-11-02361-f002:**
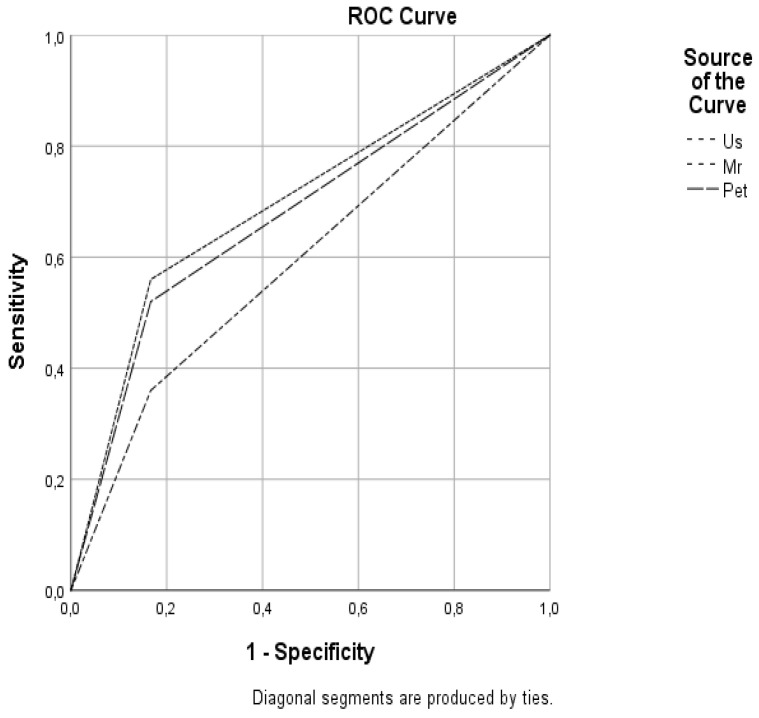
Receiver operating characteristic analysis curves of the US, MRI, and F18-FDG-PET/CT.

**Figure 3 diagnostics-11-02361-f003:**
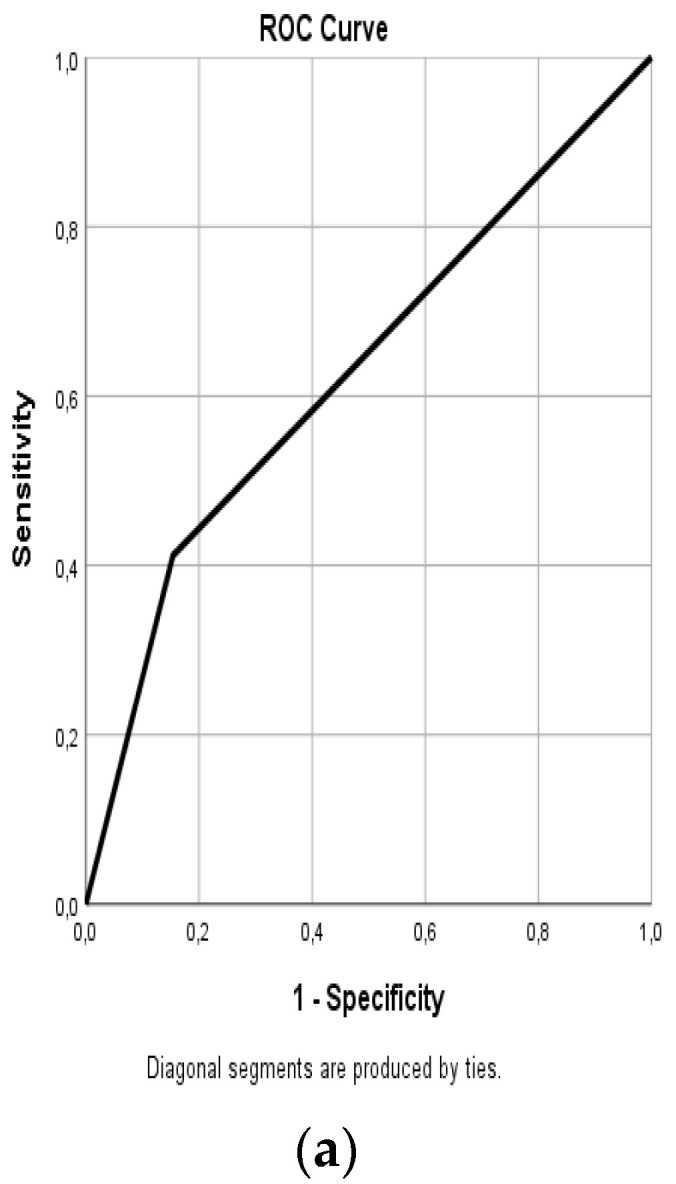
Receiver operating characteristic analysis curves of the US, MRI, and F18-FDG-PET/CT combinations. (**a**) US + MRI AUC:0.629. (**b**) USG + F18-FDG-PET/CT AUC:0.789. (**c**) F18-FDG-PET/CT + MRI AUC:0.658.

**Table 1 diagnostics-11-02361-t001:** Patient characteristics.

	Total *n* (%)	US *n* = 120(%)	MRI *n* = 48 (%)	F18-FDG-PET/CT *n* = 140 (%)
Age, year, mean, ±SD	53.8 ± 10.62	52.69 ± 10.60	51.54 ± 11.05	52.81 ± 10.58
Menopausal Status				
Postmenopausal	105 (61.4)	71 (59.2)	26 (54.2)	83 (59.3)
Premenopausal	66 (38.6)	49 (40.8)	22 (45.8)	57 (40.7)
T Status				
T1	41(24)	28 (23.3)	13(27.1)	35 (25)
T2	100(58.5)	69 (57.5)	27 (56.3)	83 (59.3)
T3	25 (14.6)	20 (16.7)	8 (16.7)	18 (12.9)
T4	5 (2.9)	3 (2.5)	0	4 (2.9)
Lymph Node Status				
N1	95(55.6)	69 (57.5)	27 (56.3)	70 (50)
N2	74 (43.3)	49 (40.8)	21 (43.8)	69 (49.3)
N3	2 (1.2)	2 (1.7)	0	1 (0.7)
Stage				
IIA	28 (16.4)	18 (15)	9 (18.8)	23 (16.4)
IIB	55 (32.2)	41 (34.2)	14 (29.2)	40 (28.6)
IIIA	82 (48)	56 (46.7)	25 (52.1)	73 (52.1)
IIIB	4 (2.3)	3 (2.5)	0	3 (2.1)
IIIC	2 (1.2)	2 (1.7)	0	1 (0.7)
Histopathology				
Invasive ductal	141 (82.5)	94 (78.3)	37 (77.1)	119 (85)
Invasive lobular	14 (8.2)	11 (9.2)	4 (8.3)	11 (7.9)
Other	16 (9.4)	15 (12.5)	7 (14.6)	10 (7.1)
ER				
Positive	132 (77.2)	92 (76.7)	41 (85.4)	108 (77.1)
Negative	39 (22.8)	28 (23.3)	7 (14.6)	32 (22.9)
PR				
Positive	123 (71.9)	87 (72.5)	37 (77.1)	102 (72.9)
Negative	48 (28.1)	33 (27.5)	11 (22.9)	38 (27.1)
HER2 Status				
Score 0	49 (28.7)	33 (27.5)	13 (27.1)	38 (27.1)
Score 1	27 (15.8)	22 (18.3)	11 (22.9)	21 (15)
Score 2	44 (25.7)	30 (25)	15 (31.3)	35 (25)
Score3	51 (29.8)	35 (19.2)	9 (18.8)	46 (32.9)
Neoadjuvant Chemotherapy				
TAC	103 (60.2)	76 (63.3)	29 (60.4)	80 (57.1)
TAC + Transzumab	41 (24)	26 (21.7)	8 (16.7)	38 (27.1)
Other	27 (15.8)	18 (15)	11 (22.9)	22 (15.7)
Operation				
Modified Radical Mastectomy	153 (89.5)	104 (86.7)	40 (83.3)	126 (90)
Total Mastectomy + SLNB	9 (5.3)	8 (6.7)	4 (8.3)	6 (4.3)
Lumpectomy + SLNB	9 (5.3)	8 (6.7)	4 (8.3)	8 (5.7)

**Table 2 diagnostics-11-02361-t002:** Imaging Techniques outcomes.

Imaging Techniques	Benign (Complete Response) *n* (%)	Malign (Incomplete Response) *n* (%)
US	70 (58,3)	50 (41,7)
MRI	33 (68,8)	15 (31,3)
F18-FDG-PET/CT	88 (62,9)	52 (37,1)

**Table 3 diagnostics-11-02361-t003:** Specificity, sensitivity, and negative and positive predictive values of imaging techniques for evaluating complete axillary response after neoadjuvant chemotherapy.

	Sensitivity (%)	Specificity (%)	Positive Predictive Value (%)	Negative Predictive Value (%)	Accuracy (%)
US	59.42	82.35	82	60	69.17
MRI	36.67	77.78	73.33	42.42	52.08
F18-FDG-PET/CT	47.50	76.67	73.08	52.27	60.00

**Table 4 diagnostics-11-02361-t004:** Specificity, sensitivity, and negative and positive predictive values of combinations of imaging techniques for evaluating complete axillary response after neoadjuvant chemotherapy.

Combinations of Imaging Techniques	Sensitivity (%)	Specificity (%)	Positive Predictive Value (%)	Negative Predictive Value (%)	Accuracy (%)
F18-FDG-PET/CT + MR	40	91.67	88.89	47.83	59.38
US + MRI	41.18	84.62	77.78	52.38	60
US + F18-FDG-PET/CT	57.89	100	100	60	74.19
US + F18-FDG-PET/CT + MRI	50	100	100	53.33	68.18

**Table 5 diagnostics-11-02361-t005:** Outcomes and mean values of previous studies.

	*n*	Sensitivity(%)			Specificity(%)			Positive Predictive Value (%)			Negative Predictive Value (%)		
		F18-FDG-PET/CT	US	MRI	F18-FDG-PET/CT	US	MRI	F18-FDG-PET/CT	US	MRI	F18-FDG-PET/CT	US	MRI
**Rousseau C.**	64	88			73			81			83		
**Peppe A.**	308		71			88							
**Morency D.**	153		52.8			78.3							
**Heiken T.J.**	272	63.2	69.8	61	84.6	58.1	58.6	85.7	71	75	61.1	56.8	42.5
**Eun N.L.**	131		81.5	87.9		70	50		71	60.7		80.8	82.5
**Croshaw R.**	61		90	86		33	79						
**Choi J.**	41		28.6	71.4		100	95.5						
**Riedinger**	47	91			86								
**Steiman J.**	135			39			88			93			26
**Schipper R.J.**	572	85	58	59	63	70	61	61	57	43	86	71	75
**Kim W.H.**	108	58		83.3	87.5		75	36.8		29.4	94.4		97.3
**Liu Q.**	382	86		65	72		88						
**You S.**	139	22	50	72	85	77	54	80	84	80	28	38	44
** Weber J.J. **	129									63.4			84.1
** Ha S.M. **	157		60	57.33	60.47		72.09						
** Means **	179.9	70.4	62.4	68.1	76.4	71.8	72.1	68.9	70.7	63.5	70.5	61.6	64.4
** Aygun **	171	4.5	59.4	36.6	76.6	82.3	77.8	73	82	73.3	52.2	60	42.4

## Data Availability

Data available on request due to privacy.

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
