# Peer review of "Efficacy of US, MRI, and F-18 FDG-PET/CT for Detecting Axillary Lymph Node Metastasis after Neoadjuvant Chemotherapy in Breast Cancer Patients"

_diagnostics, 2021, doi:10.3390/diagnostics11122361_

Round 1

Reviewer 1 Report

This is a retrospective study comparing MRI, US and FDG PET for the detection of residual axillary lymph nodes metastases in locally advanced breast cancer, treated with neoadjuvant chemotherapy.  The authors showed that US had the highest accuracy, while the combination of US and FDG PET presented a sensitivity of 50% and a specificity of 100%.

The paper is interesting, well written, easy to read and focused on a relatively few explored topic. Here some considerations:

  • page 3, line 138, the authors wrote: "when no FDG uptake could be visually identified, the SUV was recorded as zero, while lymph nodes with a maximum SUV above zero were considered metastatic". It is not clear to me. Why did the authors choose such a  threshold? Did they find other similar papers adopting the same approach? Did they also consider the pattern of uptake (focal or diffuse)? Please elaborate.
  • While the overall number of performed FDG PET and US exams are sufficiently comparable (140 and 120, respectively), the number of performed MRIs is meaningfully lower (i.e. 48). This discrepancy might have determined a bias in comparison. What do the authors think about this? I suggest to add a sentence on this limitation in Discussion.
  • The authors did not specify if they use an analog or digital PET/CT device (I assume they used an analog one, which is extremely more common). The type of PET/CT device utilized for the study should be mentioned. Furthermore, I suggest to add in Discussion a brief sentence on the potential of digital PET/CT, characterized by superior resolution and sensitivity, in the field of interest.
  • I suggest to insert an illustrative diagnostic image. 

Author Response

1-The section related to FDGPET in the Material-Method has been revised. Thank you for this important review. The number of separate FDG-avid lesion in the breast and regional lymph node stations was visually assessed after anatomical localization. Any uptake in axilla which could be mapped to a lymph node was considered abnormal and was rated for metastasis. A given PET focus was rated as positive in a small lymph node rather than in a large lymph node, accounting for partial volume effects. Region of interest  was placed over the most intense area of 18F-FDG accumulation in axillary lymph nodes. There were some studies that there evaluated FDG uptake with this method (Güney IB et al. A retrospective comparative study of ultrasonography, contrast-enhanced MRI and  18F-FDG PET/CT for preoperative detection of axillary lymph node metastasis in breast cancer patients. Ann Ital Chir 2020; 91)

2-The fact that the number of MRI performed on the patients included in our study was lower than PET and US may be a shortcoming of our study. The number of MRI was low, as we preferred breast MRI only in young patients and patients with a dense breast pattern. However, despite the small number of patients, we wanted to use MRI in our study in the evaluation of axillary lymph nodes.

3- Thank you for this comment. This sentence has been inserted in material methods section: An analog PET/CT device was used in the study. Whole body 18F-FDG PET/CT scans were obtained on a Biograph PET/CT system (Siemens Molecular Imaging, Hoffman Estates, IL, USA). The system consists of a full ring dedicated PET and a 2-slice spiral CT.

4- We added a metastatic lymph node image from PETCT.

Reviewer 2 Report

The authors assess the diagnostic accuracy of axillary imaging modalities (US, MRI, PET) after neoadjuvant chemotherapy for women with node positive breast cancer.

The topic is an important one. However, the manuscript as currently written has shortcomings.

  1. It appears that the study including individuals with "clinically" positive nodes that were not confirmed by biopsy, as well as individuals who had biopsy confirmed disease.

These are disparate groups and should not be combined.

  1. Many are now indicating that all nodes that are biopsied due to a suspicion of nodal spread from the breast should be clipped (e.g. Boughey, Ann Surg Oncol 2019, 26:3794f). It is not clear what fraction of the participants in this study had their biopsied nodes clipped, and if clipped, what fraction of the clipped nodes were correctly imaged and retrieved. The clip, of course, can be seen by some forms of imaging which can skew imaging findings, so that must be addressed.
  2. Details of how the post NAC nodes were retrieved are inadequate. Were they retrieved by SLN, ALND, or some combination? These are different groups and need to be treated separately.

Author Response

1,2- We included patients with clinically positive lymph nodes and biopsy-proven patients at the time of diagnosis. In this study, we did not evaluate the response to NACT in breast cancer patients, we only demonstrate the efficacy of imaging modalities after NACT(pre-operatively)  by comparing it with histopathology. Therefore, we do not think that biopsy at the time of diagnosis is necessary. The lack of clipping of lymph nodes and all patients had no histopathologically proven lymph node metastasis at the time of diagnosis  in our study may be a deficiency, especially in patients who underwent SLNB, but SLNB was performed in only 10.6% of the patients, axillary dissection was performed in approximately 90% of the patients, included in our study, so we do not think that there is a serious shortcoming

3- How the lymph nodes were removed after NACT (which operation was performed) is shown in Table-1. There are 3 groups of operations in our study, modified radical mastectomy(total mastectomy+axillary dissection 89.6%), total mastectomy+SLNB (5.3%), lumpectomy+SLNB (5.3%). Patients with positive SLNB results and who underwent axillary dissection were included in the axillary dissection group.

Round 2

Reviewer 2 Report

As previously mentioned, I do not feel it is appropriate to combine clinically positive and biopsy confirmed positive LN. The number of patients with clipped vs. not clipped nodes remains important, should be listed and analyzed separately. Finally, as mentioned, ALND vs. SLN retrieved nodes should be analyzed separately in the determination of imaging modality efficacy.

Author Response

Axillary biopsy was not required in patients who were clinically considered to have axillary metastases in the previous guidelines available at the time of the study. Likewise, clipping of axillary lymph nodes with metastasis was not routinely recommended because sentinel lymph node or targeted biopsy was not routine after neoadjuvant at that time (this issue still contains controversial aspects).

Therefore, complete axillary dissection (levels 1 and 2) was performed in a significant part of the patients who received neoadjuvant therapy in our study. Therefore, when we look at our study from that perspective, we do not think that this is a limitation. On the contrary, we think it is an advantage. Because the axillary dissection performed in most cases gives the gold standard information about the axilla and will prevent the evaluation errors that may arise due to the nature of the sentinel lymph node biopsy.

We understand the reservation of reviewer 2. However, because of the reasons mentioned above, our study is precious and gives lots of information for future researches and makes a critical status determination. Therefore, we think that a proposed separation will not change our conclusion. Readers already can easily see this separation in the tables of this manuscript.

We regret to inform you that it would not be fair if the study were not published just for this reason and to ignore other important information that it clarifies about clinical applications. We think that this issue should be left to the understanding of the readers.

On the other hand, to make things easy and respect for the reviewers:

We add the following information to the discussion section in line with the recommendations of reviewer 2 to our manuscript:

Unfortunately, our study has some limitations. Axillary biopsy was not required in patients who were clinically considered to have axillary metastases in the previous guidelines available at the time of the study. Likewise, clipping of axillary lymph nodes with metastasis was not routinely recommended because sentinel lymph node or targeted biopsy was not routine after neoadjuvant treatment at that time. Additionally, we separated the patients who had axillary dissection and sentinel lymph node biopsy in Tables 1 and 2. On the other hand, in today's perspective, the lack of clipping in patients with sentinel lymph nodes and the absence of axillary biopsies in patients with clinically axillary metastases are limitations and shortcomings of this study.

Round 3

Reviewer 2 Report

Analysis of the groups should be separated, not combined, as mentioned in the prior review. It does not appear that this was done. 

Author Response

Dear reviewer;

 we added a paragraph about the outcomes of the patients who were performed only ALND to the "results" and "Discussion"section